# Hospital Resources May Be an Important Aspect of Mortality Rate among Critically Ill Patients with COVID-19: The Paradigm of Greece

**DOI:** 10.3390/jcm9113730

**Published:** 2020-11-20

**Authors:** Christina Routsi, Eleni Magira, Stelios Kokkoris, Ilias Siembos, Charikleia Vrettou, Dimitris Zervakis, Eleni Ischaki, Sotiris Malahias, Ioanna Sigala, Andreas Asimakos, Theodora Daidou, Panagiotis Kaltsas, Evangelia Douka, Adamandia Sotiriou, Vassiliki Markaki, Prodromos Temberikidis, Apostolos Koroneos, Panagiotis Politis, Zafiria Mastora, Efrosini Dima, Theodoros Tsoutsouras, Ioannis Papahatzakis, Panagiota Gioni, Athina Strilakou, Aikaterini Maragouti, Eleftheria Mizi, Ageliki Kanavou, Aikaterini Sarri, Evdokia Gavrielatou, Spyros Mentzelopoulos, Ioannis Kalomenidis, Vassilios Papastamopoulos, Anastasia Kotanidou, Spyros Zakynthinos

**Affiliations:** 1First Department of Intensive Care Medicine, School of Medicine, National and Kapodistrian University of Athens, ‘Evangelismos’ Hospital, 45–47 Ipsilandou St, GR-10675 Athens, Greece; chroutsi@hotmail.com (C.R.); elmagira@med.uoa.gr (E.M.); skokkoris2003@yahoo.gr (S.K.); ils2007@med.cornell.edu (I.S.); vrettou@hotmai.com (C.V.); dzervakis@hotmail.com (D.Z.); eischaki@yahoo.gr (E.I.); sotmalachias@gmail.com (S.M.); giannasig@yahoo.com (I.S.); silverako@gmail.com (A.A.); theodorantaidou@yahoo.com (T.D.); nkaltsas@hotmail.com (P.K.); ldoukagr@yahoo.com (E.D.); mandiasotiriou@hotmail.com (A.S.); vmarkaki@keystone.gr (V.M.); makistemper@yahoo.gr (P.T.); koroneos@hotmail.com (A.K.); politispa@yahoo.com (P.P.); zefimast@yahoo.gr (Z.M.); efi_dima@yahoo.gr (E.D.); theo.tsoutsouras@gmail.com (T.T.); yiannispapahatzakis@gmail.com (I.P.); pgioni@yahoo.com (P.G.); athinastrilakou@yahoo.com (A.S.); katrinmara@hotmail.com (A.M.); eleftheria.mizi@yahoo.com (E.M.); agkanavou@yahoo.com (A.K.); katsarri5@hotmail.com (A.S.); ev.gavrielatou@gmail.com (E.G.); sdmen@med.uoa.gr (S.M.); ikalom@med.uoa.gr (I.K.); kotanidou@med.uoa.gr (A.K.); 2Fifth Department of Internal Medicine, Unit for Infectious Diseases, ‘Evangelismos’ Hospital, 45–47 Ipsilandou Street, GR-10675 Athens, Greece; vpapastam@yahoo.com

**Keywords:** coronavirus, coronavirus disease 2019, COVID-19, critical care, acute hypoxemic respiratory failure, mortality, acute respiratory distress syndrome, ARDS, sequential organ failure assessment, SOFA

## Abstract

For critically ill patients with coronavirus disease 2019 (COVID-19) who require intensive care unit (ICU) admission, extremely high mortality rates (even 97%) have been reported. We hypothesized that overburdened hospital resources by the extent of the pandemic rather than the disease per se might play an important role on unfavorable prognosis. We sought to determine the outcome of such patients admitted to the general ICUs of a hospital with sufficient resources. We performed a prospective observational study of adult patients with COVID-19 consecutively admitted to COVID—designated ICUs at Evangelismos Hospital, Athens, Greece. Among 50 patients, ICU and hospital mortality was 32% (16/50). Median PaO_2_/FiO_2_ was 121 mmHg (interquartile range (IQR), 86–171 mmHg) and most patients had moderate or severe acute respiratory distress syndrome (ARDS). Hospital resources may be an important aspect of mortality rates, since severely ill COVID-19 patients with moderate and severe ARDS may have understandable mortality, provided that they are admitted to general ICUs without limitations on hospital resources.

## 1. Introduction

The outbreak of coronavirus disease 2019 (COVID-19) emerged in China in December 2019 [1,2] and rapidly spread worldwide. The major clinical complication in patients with COVID-19 is acute respiratory distress syndrome (ARDS) requiring intensive care unit (ICU) admission. For those patients needing care in an ICU, mortality rates as high as 49 to 97% have been reported [3,4,5,6,7,8,9,10]. However, lower mortality rates (30.9% for all patients admitted to the ICU and 35.7% for intubated patients) have been reported in areas where ICU capacity enabled the timely admission of all COVID-19 patients requiring critical care to a traditional ICU [11].

In Greece, the first confirmed case of COVID-19 was detected on 26 February 2020 and on 11 March the government took restrictive measures to curb the spread of the disease, including closure of all educational institutions throughout the country, followed by total lockdown. Thereafter, a successful response to the outbreak by slowing the spread of COVID-19 was observed and the number of patients with confirmed COVID-19 and of those who required hospital and ICU admission was limited, thus not overwhelming hospital resources [12,13].

We hypothesized that overburdened hospital resources by the extent of the pandemic rather than the disease per se might play an important role on the unfavorable prognosis of critically ill patients with COVID-19 requiring ICU admission. Therefore, we considered that investigating the outcome of critically ill patients with COVID-19 admitted to the general ICUs of a big referral for COVID-19 tertiary-care hospitals with sufficient resources would provide valuable insight.

## 2. Materials and Methods

### 2.1. Patients and Setting

This is a prospective observational study of all adult patients with COVID-19 consecutively admitted to four COVID-19 designated ICUs at Evangelismos Hospital, Athens, Greece, from 11 March to 27 April 2020. COVID-19 status was based upon positive severe acute respiratory syndrome coronavirus 2 real-time reverse transcriptase–polymerase chain reaction assay of nasopharyngeal-swab specimens. The study was approved by the hospital institutional review boards. Informed consent was waived.

During the study period, all patients with COVID-19 who required critical care were timely admitted to a COVID-ICU due to adequate ICU capacity. Additionally, all patients admitted to COVID-ICUs were treated by standard (i.e., pre-COVID) multidisciplinary ICU care teams with usual ICU staffing ratios. There was no lack of ventilators or other respiratory support devices, dialysis machines, medications, or other critical care supplies including personal protective equipment. Although there was an institutional recommendation against the routine use of non-invasive positive pressure mechanical ventilation during the COVID-19 pandemic for patients with COVID-19, mainly based on the Surviving Sepsis Campaign Guidelines [14], the routine use of high-flow nasal cannula oxygen therapy was not limited.

Clinical management was left at the discretion of the critical-care trained attending physician. The management of mechanically ventilated patients was provided according to ARDS treatment guidelines [14,15], including prone position ventilation and conservative fluid management. Briefly, tidal volumes of approximately 6 mL/kg predicted body weight and a respiratory rate adjusted to maintain arterial pH above 7.30 were applied, targeting plateau pressure of less than 30 cmH_2_O. Positive end-expiratory pressure (PEEP) was usually set according to the lower PEEP/higher FiO_2_ARDS network table [15], titrating for the best tidal respiratory system compliance.

An historical group of 64 patients with ARDS without COVID-19, who were included in a previous publication from our department, was used as control group [16]. Most of these patients had pneumonia as the cause of ARDS; all patients were ventilated with lung-protective conventional mechanical ventilation (CMV), constituting the CMV group of this study [16], and were managed according to the ARDS treatment guidelines [14,15] by the same medical staff involved in the current study.

### 2.2. Data Collection and Definitions

Data were collected through to 16 June 2020. All patient data, clinical and laboratory, as well as ventilator settings and measurements were prospectively collected along with the ongoing pandemic. ARDS was defined according to the Berlin Definition criteria [17]. Severity of illness was assessed by the Acute Physiology and Chronic Health Evaluation (APACHE) II [18] and Sequential Organ Failure Assessment (SOFA) [19] scores. The severity of acute pulmonary damage in mechanically ventilated patients was graded by using the Lung Injury Score [20]. Respiratory system static compliance was computed as the tidal volume divided by the difference between plateau pressure and total PEEP. The last difference constituted the driving pressure. The evolution of organ dysfunction during the ICU stay was evaluated by calculating the SOFA score at admission and on ICU days 3, 5, 10, 14 and 21.

### 2.3. Statistical Analysis

Quantitative data are reported as median and interquartile range (IQR). Non-parametric statistics were applied. Comparisons between patients who survived versus those who died in the ICU were performed by using the Mann–Whitney U test. Differences between these two groups of patients in qualitative variables were assessed by Chi-square or Fisher’s exact test when appropriate. Differences in group data for related samples were evaluated by the Friedman’s ANOVA by ranks; when significant differences were found, post-hoc comparisons were performed by using the Wilcoxon matched-pair test. The SPSS statistical program (version 10, Chicago, IL, USA) was used for data analysis. Statistical significance was defined as a two-tailed *p* value of <0.05.

## 3. Results

### 3.1. Patient Characteristics and ICU Admission

During the study period, 50 adults with COVID-19 infection were critically ill and admitted to the ICUs of our institution (Figure 1a). Demographics and data on ICU admission are summarized in Table 1. The median patient age was 64 years (IQR, 58–72) and 24 patients (48.0%) were 65 years or older. The majority of patients were Caucasian (47 (94.0%)). Hypertension was the most common comorbidity (14 (28.0%)), followed by diabetes (9 (18.0%)). Five patients (10.0%) had hematologic malignancy, whereas 18 patients (36.0%) had at least two comorbidities.

The median time from symptom onset to hospital admission was seven days (IQR, 5.0–8.5 days). Twenty-three patients (46%) were admitted to the ICU within 24 h after hospital admission. The remaining 27 patients (54%) had median hospitalization times in general wards prior to ICU admission two days (IQR, 1–4 days)

On ICU admission, the median APACHE II score was 12 (IQR, 8–17), and the median SOFA score was 7 (IQR, 3–9). Bilateral lung infiltrates were shown in 49 patients (98%), whereas unilateral lung infiltrates was shown in only one patient (2%). Computed tomography was performed in 36 patients; all of them had bilateral patchy shadows and ground-glass opacities with multilobe involvement. The median PaO_2_/FiO_2_ was 121 mmHg (IQR, 86–171 mmHg). Thirty-nine patients (78%), all under invasive mechanical ventilation, were receiving noradrenaline for circulatory support. In none of the patients was the respiratory viral panel positive for a different viral infection or the respiratory sample positive for bacteria.

Invasive mechanical ventilation was applied in 41 patients (82%) (Figure 1b). Among 29 patients who were intubated in our ICU, in 22 the indication for intubation was severe hypoxemia combined with dyspnea, in four it was hypoxemia without severe dyspnea and in the remaining three patients the main indication was shock combined with decreased consciousness. The decision to intubate was always made by the attending physician. The remaining 12 patients were admitted to our ICU already intubated and invasively ventilated from other hospitals; therefore we are unable to identify the indication for their intubation and who decided it. For the patients intubated in our ICU, on the day of intubation median static, respiratory system compliance was 40 mL/cmH_2_O (IQR, 32–50 mL/cmH_2_O) and median Lung Injury Score was 2.7 (IQR, 2.5–3.2) (ARDS is identified when Lung Injury Score is higher than 2.5 [20]).

### 3.2. Interventions in the ICU

Interventions in the ICU are included in Table 2. A total of 14 patients (28%) received high-flow nasal cannula, whereas only 2 patients (4%) were treated with non-invasive mechanical ventilation. All patients who did not receive invasive mechanical ventilation (*n* = 9) were managed with high-flow nasal cannula oxygen therapy with 60 L/min air-flow and high FiO_2_; two of these patients were also managed with non-invasive mechanical ventilation. High-flow nasal cannula oxygen therapy and noninvasive mechanical ventilation were applied in ICU rooms with negative atmospheric pressure in order to minimize staff exposure to the virus. Among those who received invasive mechanical ventilation (*n* = 41), the median duration of mechanical ventilation was 13 days (IQR, 9–33 days), while 26 (52%) required neuromuscular blockade therapy for at least 24 h, and prone position was used as rescue therapy in 6 patients (12%). Thirty-four patients (68%) required noradrenaline as vasopressor support for shock and 13 patients (26%) needed continuous renal replacement therapy. Most of the patients received hydroxychloroquine for at least 5 days.

### 3.3. ICU Outcomes

ICU outcomes are displayed in Figure 1 and are summarized in Table 2. Due to data censoring on 16 June 2020, median patient follow-up was 50 days (range 29–88) (Table 2). Among 50 patients admitted in the ICU, 16 died (32.0%), 33 patients were discharged alive from the ICU (66.0%), and 1 patient remained in the ICU (2.0%) still receiving invasive mechanical ventilation; hospital mortality was 32.0% (16/50) (Figure 1a), whereas 28-day mortality was 24% (12/50) (Table 2). Among 41 patients who received invasive mechanical ventilation, ICU and hospital mortality was 39.0% (16/41) (Figure 1b); 24 patients were discharged alive from the ICU (58.6%), 17 of them after a successful extubation and the remaining 7 patients with a tracheostomy in place (Table 2). Twenty-eight-day mortality was 29% (12/41) (Table 2). Among 9 patients who did not receive invasive mechanical ventilation, ICU and hospital mortality was 0% (0/9) (*p* = 0.02 versus the mortality of patients who received invasive mechanical ventilation).

Patients who died had a higher proportion of current malignancies, and on ICU admission, had higher APACHE II and SOFA scores, higher blood troponin T and lactate values, and were receiving noradrenaline for vasopressor circulatory support more frequently than those who survived (Table 1). Notably, among the patients who died, three (19%) had suffered a cardiac arrest during endotracheal intubation. Compared to those who survived, patients who died in the ICU rarely received high-flow nasal cannula oxygen therapy, and more commonly required invasive mechanical ventilation, noradrenaline infusion and continuous renal replacement therapy for renal failure management (Table 2). Duration of hospital stay was longer in patients who survived compared to those who died in ICU (Table 2).

Patients who did not receive invasive mechanical ventilation (*n* = 9) were less sick than those who received invasive mechanical ventilation (*n* = 41) as indicated by the lower APACHE II and SOFA scores in the former group ((mean ± SD) 7.4 ± 3.0 and 2.4 ± 0.7, respectively) versus (14.6 ± 6.9 and 7.9 ± 3.3, respectively) the latter group of patients. However, patients who did not receive invasive mechanical ventilation had lower PaO_2_, PaO_2_/FiO_2_, and PaCO_2_ (90.7 ± 17.6 mmHg, 100.3 ± 17.9 mmHg, and 31.5 ± 3.4 mmHg, respectively) compared with those who received invasive mechanical ventilation (115.8 ± 42.8 mmHg, 147.5 ± 74.5 mmHg, 42.6 ± 10.2 mmHg, respectively).

SOFA scores in patients who died were higher than in those who survived not only on ICU admission, but also on days 3–21 (*p* < 0.01 for all days) (Figure 2). This difference could be attributed to higher cardiovascular and renal components of SOFA scores in patients who died compared to those who survived ICU. SOFA scores in patients who survived were not significantly different from ICU admission to day 21. In contrast, SOFA scores in patients who died did differ in the course of their ICU stay; post-hoc analysis showed that their SOFA scores on days 10, 15 and 21 were all significantly higher than on admission, day 3 or day 5 (*p* < 0.05 for all days) (Figure 2). This difference was mainly due to the higher renal component of SOFA scores. The primary cause of death of all patients was multi-organ failure, most commonly due to sepsis; none died from refractory hypoxemia, neurologic dysfunction or withdrawal of life support (18). Thrombotic events (i.e., deep venous thrombosis) occurred in three patients (6%) (two patients survived and one died); in one patient who eventually survived, deep venous thrombosis was complicated by an intermediate-risk pulmonary embolism. ICU-acquired bacteremia was detected in 12 patients who survived (36%) versus 4 patients who died (25%) (*p* = 0.637). The median number of bacteremia in patients who survived was 2.5 (IQR, 1.0–4.5) versus 1.5 (IQR, 1.0–4.0) in patients who died (*p* = 0.770). *Klebsiellapneumoniae*, *Acinetobacterbaumannii* and *Enterococcus faecalis* were the most frequent bacteria, and were isolated in 6 (18%), 6 (18%) and 5 patients (15%), respectively, in patients who survived compared to 3 (19%), 2 (13%) and 2 patients (13%), respectively, in patients who died (*p* = 0.729–0.926).

### 3.4. The Control Group 

Table 3 displays the comparison of the historical control group of patients with ARDS without COVID-19 [16] with the intubated and invasively ventilated patients with COVID-19 of the current study; all these latter patients fulfilled the criteria of ARDS [17]. Compared to ARDS with COVID-19, patients with ARDS without COVID-19 in the control group were younger, had higher APACHE II and SOFA scores, higher LIS, and lower PaO_2_/FiO_2_ ratios and respiratory system static compliance. Hospital mortality of ARDS without COVID-19 was 64.1% compared with 39.0% of ARDS with COVID-19.

## 4. Discussion

The main findings of the present study were: (i) ICU and hospital mortality were 32%, 39% and 0% for the entire cohort of patients admitted to the ICU, for those who received invasive mechanical ventilation, and for those who did not receive invasive mechanical ventilation, respectively; (ii) 28-day mortality was 24% and 29% for the whole group of patients admitted to the ICU and for those who required invasive mechanical ventilation, respectively; (iii) patients who eventually died already had an increased risk of death even on ICU admission, as suggested by the high values of APACHE II and SOFA scores, the presence of current malignancy and occurrence of cardiac arrest, and the generalized need for circulatory support with noradrenaline; (iv) the primary cause of death of all patients was multi-organ failure most commonly due to sepsis, whereas none died from refractory hypoxemia, neurologic dysfunction or withdrawal of life support; and (v) hospital stay was longer in patients who survived, frequently complicated by bacteremia.

Mortality rates of ICU patients with COVID-19 in the present study were substantially lower than those reported in studies from countries with a great burden of the COVID-19 pandemic (China, USA and Italy), resulting in an overburdened medical system and insufficient hospital resources. Indeed, ICU and hospital mortality of 32% and 39% for overall ICU admissions and for intubated patients, respectively, observed in the present study, were considerably lower than those of earlier studies in ICU patients with COVID-19, reporting overall mortality rates 49–62% [3,8,9] and mortality rates as high as 66–97% among patients requiring invasive mechanical ventilation [8,21,22] in China. Similarly, the mortality rates for our patients were substantially lower than those of subsequent reports from USA [5,6,7], showing mortality rates for patients requiring invasive mechanical ventilation ranging from 52% [6] and 67% [7] in the Seattle region to 88.1% in the New York area [5], as well as from Italy, demonstrating an overall hospital mortality of 53.4% for patients admitted to ICU [23]. Therefore, an overburdened medical system with subsequent insufficient hospital resources was associated with the extent of the pandemic rather than the disease per se and might play an important role on the unfavorable prognosis of critically ill patients with COVID-19 requiring ICU admission. In our study, all critically ill patients with COVID-19 were admitted to the ICU in time, because overwhelming stress on the healthcare system did not occur in our country [12,13]. All these patients were admitted to preexisting multidisciplinary ICUs, were cared for by critical care teams with experience in the management of patients with severe acute respiratory failure at standard staffing ratios, and received full intensive care support, including renal replacement therapy. Another factor that could have played a role is that the onset and peak of the COVID-19 pandemic in Greece occurred later in Greece than in many regions from earlier reports, including neighboring Italy [23]. This delay gave time to make organizational arrangements, purchase equipment, train personnel, make consensus-driven clinical protocols, and distribute supplies across the healthcare system.

Most of our patients had moderate and severe ARDS, according to the ARDS Berlin Definition [17]; the definition of ARDS was also fulfilled in all our mechanically ventilated patients on the day of intubation [17,20] (Table 1 and Table 3). These latter patients had hospital mortality rates of 39.0% compared to 64.1% in the historical control group of ARDS without COVID-19 (Table 3). Patients of the control group had more severe disease as indicated by the higher APACHE II and SOFA scores, and lower PaO_2_/FiO_2_ ratios and respiratory system static compliance. Nevertheless, our data, as well as that of Auld et al. [11], provide evidence that mortality rates of severely ill COVID-19 patients with ARDS may be comparable or even lower to those reported in ARDS of different etiologies [24,25], including Influenza A [26,27]. Indeed, hospital mortality for patients with moderate ARDS was 40.3% and for those with severe ARDS it was 46.1% (24), whereas the overall pooled mortality rates of all ARDS studies included in a comprehensive literature review was 43% [25]. Moreover, mortality rates were 41.4% [26] and 46% [27] among ICU patients with Influenza A pneumonia, most of whom had ARDS.

Of note, our patients with COVID-19-related ARDS had a mean static compliance 35% higher than the mean static compliance of the group of patients with ARDS without COVID-19 (Table 3). This finding is compatible with those of several studies which have suggested that patients with COVID-19-related ARDS have a markedly higher lung compliance than patients with ARDS unrelated to COVID-19 [28,29,30]. However, just like in a recent study [31], the values of static compliance in our patients with COVID-19-related ARDS overlap to a great extent with those in patients with ARDS unrelated to COVID-19 (Table 3).

Lung disease on ICU admission was not worse in patients who died than in those who survived as suggested by lack of any difference in blood oxygenation, respiratory system compliance and Lung Injury Score values. There was also no difference in PEEP, plateau pressure and driving pressure levels (Table 1). Moreover, lung disease severity on admission and ICU on days 3–21 did not contribute to the higher SOFA scores in patients who died compared to those who survived (Figure 2) and nobody died from refractory hypoxemia. The higher SOFA scores on admission and on ICU days 3–21 (Figure 2) were mainly due to the higher cardiovascular and renal components of SOFA scores. Not surprisingly, patients who died compared to those who survived more commonly needed noradrenaline infusion for circulatory support on ICU admission (Table 1), and required noradrenaline use for shock treatment and continuous renal replacement therapy for renal failure management during their ICU stay (Table 2).

We acknowledge the main limitation of this study, i.e., we reported findings and outcomes from a single center and included a rather limited number of patients compared to hundreds or thousands of patients incorporated in many studies from other countries with thousands of victims of the COVID-19 pandemic [3,4,5,10,21,23,32]. However, our study included all consecutive patients admitted to four COVID-designated ICUs of the biggest referral center for COVID-19 in Greece and our study sample represents about 25% of the total number of patients admitted to Greek ICUs [12]. Due to the relatively limited spread of the pandemic in Greece, our medical system was not overwhelmed, and this fact generated the purpose of the present study. Therefore, our findings might not be generalizable to different populations and medical systems, but might be relevant to other countries where the pandemic did not overwhelm the health system capacity.

## 5. Conclusions

Hospital resources may be an important aspect of mortality rates among critically ill patients with COVID-19, since overburdened hospital resources by the extent of the pandemic rather than the disease per se might play an important role on unfavorable prognosis of patients requiring ICU admission. Severely ill COVID-19 patients with moderate and severe ARDS may have equal or even lower mortality rates compared to ARDS attributed to other causes, given that they are admitted to general ICUs with experienced and sufficient staff without limitations in hospital resources where established ARDS therapies [14,15], including low tidal volume and conservative fluid management, are used.

## Figures and Tables

**Figure 1 jcm-09-03730-f001:**
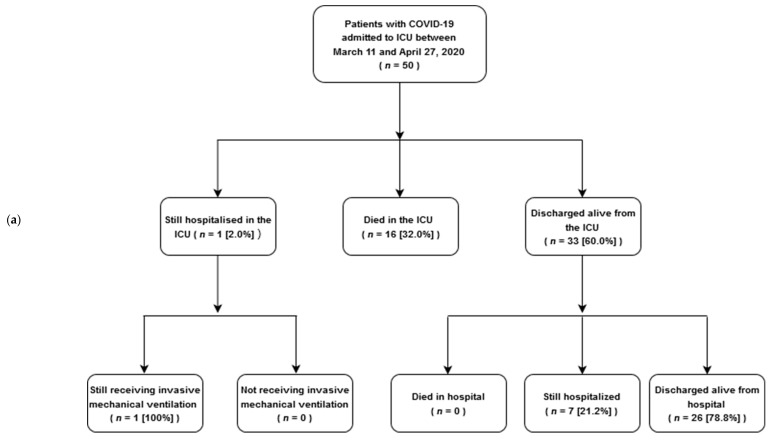
Flow diagram for study patients who were admitted to COVID-19 designated ICUs (**a**), and received invasive mechanical ventilation (**b**). Abbreviations: COVID-19 = coronavirus disease 2019, ICU = intensive care unit.

**Figure 2 jcm-09-03730-f002:**
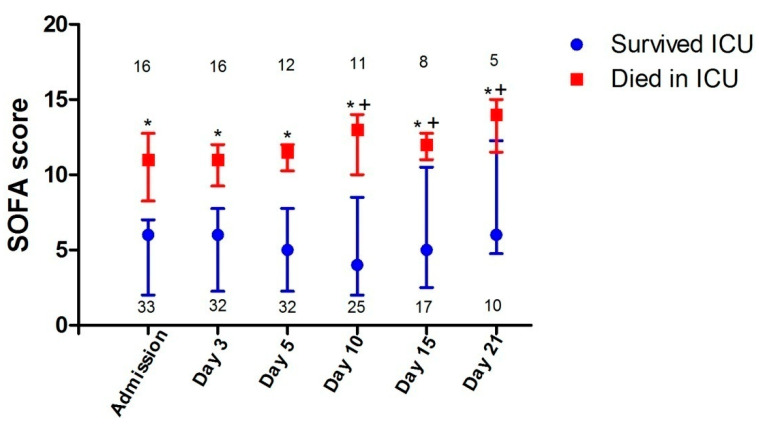
Serial SOFA scores of patients admitted to the ICU and either survived or died in ICU. Days represent ICU days. Values are medians and error bars are interquartile ranges. Numbers represent the number (*n*) of patients on each day. Asterisks (*) denote significant differences between those who died and those who survived on the same ICU day (*p* < 0.01; Mann–Whitney U test), whereas crosses (+) denote significant differences in patients who died between SOFA scores on days 10, 15 and 21 and SOFA scores on admission, day 3 or day 5 (*p* < 0.05; Wilcoxon matched-pair test). Abbreviations: SOFA = Sequential Organ Failure Assessment, ICU = intensive care unit.

**Table 1 jcm-09-03730-t001:** Demographics and clinical and laboratory data on ICU admission of patients with COVID-19 *^a^*.

	All (*n* = 50) *^b^*	Survived ICU (*n* = 33)	Died in ICU (*n* = 16)	*p ^c^*
Age, years (median (IQR))	64 (58–72)	61 (55–71)	70 (60–77)	0.065
≤64	26 (52.0)	20 (60.6)	5 (31.3)	0.054
≥65	24 (48.0)	13 (39.4)	11 (68.7)	
Gender, male	38 (76.0)	25 (75.8)	12 (75.0)	0.954
Race				
Caucasian	47 (94.0)	30 (90.9)	16 (100)	0.542
Asian	3 (6.0)	3 (9.1)	0 (0)	
Comorbidity				
Hypertension	14 (28.0)	9 (27.3)	4 (25.0)	0.866
Diabetes mellitus	9 (18.0)	6 (18.2)	3 (18.8)	0.962
Cardiovascular *^d^*	6 (12.0)	3 (9.1)	3 (18.8)	0.333
Malignancy *^e^*	5 (10.0)	1 (3.0)	4 (25.0)	**0.034**
Obesity *^f^*	5 (10.0)	3 (9.1)	2 (12.5)	0.712
Chronic lung disease	4 (8.0)	2 (6.1)	2 (12.5)	0.440
Chronic renal failure	1 (2.0)	0 (0)	1 (6.3)	0.327
Comorbidities (≥2)	18 (36.0)	11 (33.3)	7 (43.8)	0.478
Time from symptom onset to ICU admission, days (median (IQR))	9 (8–11)	10 (8–12)	8 (8–10.5)	0.352
APACHE II score, median (IQR)	12 (8–17)	9 (6–12)	20 (17–26)	**<0.001**
SOFA score, median (IQR)	7 (3–9)	6 (2–7)	11 (9–13)	**<0.001**
Laboratory data, median (IQR)				
Leukocyte count, ×10^9^/L	7.6 (5.9–11.2)	7.3 (5.9–10.8)	9.5 (6.0–17.3)	0.386
Lymphocyte count, ×10^9^/L	0.88 (0.61–1.13)	0.88 (0.65–1.13)	0.93(0.48–1.08)	0.855
Platelet count, ×10^9^/L	233 (193–318)	233 (200–322)	263 (211–329)	0.553
C-reactive protein, mg/dL *^g^*	14.2 (7.7–24.6)	13.6 (7.7–20.3)	21.7 (9.6–31.9)	0.092
Procalcitonine, ng/mL *^h^*	0.43 (0.12–0.94	0.34 (0.15–0.66)	1.41 (0.11–1.52)	0.503
Ferritin, μg/L	835 (353–1882)	835 (576–1710)	1509 (479–1740)	0.345
D-dimer, μg/mL *^i^*	1.23 (0.52–2.46)	1.30 (0.59–2.43)	1.17 (0.45–2.17)	0.987
Lactate dehydrogenase, IU/L	471 (403–611)	454 (363–604)	509 (428–574)	0.316
Albumin, g/dL	3.2 (3.0–3.6)	3.3 (3.1–3.7)	3.0 (2.8–3.4)	**0.038**
Creatinine, mg/dL	0.9 (0.7–1.1)	0.8 (0.7–1.0)	1.1 (0.7–1.8)	0.123
TroponinΤ, pg/mL *^j^*	15 (10–52)	12 (8–20)	31 (14–85)	**0.012**
Lactate, mmol/L	1.3 (1.0–1.8)	1.0 (0.9–1.3)	1.8 (1.3–2.1)	**<0.001**
Bilateral infiltrates on chest x-ray/CT	49 (98)	32 (97)	16 (100)	1.000
Arterial blood gases, median (IQR)				
PaO_2_, mmHg	100 (83–130)	101 (86–121)	99 (75–155)	0.977
PaO_2_/FiO_2_, mmHg	121 (86–171)	119 (89–157)	123 (75–243)	0.790
PaCO_2_, mmHg	40 (33–45)	36 (29–42)	44 (42–50)	**0.002**
pH	7.39 (7.32–7.44)	7.42 (7.35–7.45)	7.30 (7.25–7.36)	**0.006**
Respiratory parameters on day of intubation, median (IQR) *^k^*				
PEEP, cmH_2_O	14 (12–16)	14 (12–17)	13 (10–16)	0.436
Plateau pressure, cmH_2_O	27 (25–29)	27 (26–29)	25 (24–29)	0.413
Driving pressure, cmH_2_O	13 (11–15)	12 (10–14)	13 (12–15)	0.384
Static compliance, mL/cmH_2_O	40 (32–50)	40 (33–50)	39 (33–42)	0.683
LIS	2.7 (2.5–3.2)	2.7 (2.5–3.2)	2.8 (2.5–3.5)	0.730
Noradrenaline	39 (78)	21 (64)	16 (100)	**0.004**

ICU, intensive care unit; IQR, interquartile range; BMI, body mass index; APACHE, Acute Physiology and Chronic Health Evaluation [18]; SOFA, Sequential Organ Failure Assessment [19]; PaO_2_, partial pressure of arterial oxygen; FiO_2_, fraction of inspired oxygen; PaCO_2_, partial pressure of arterial carbon dioxide; CT, computed tomography; PEEP, positive end-expiratory pressure; LIS, Lung Injury Score [20]. *^a^* Data are expressed as *n* (%) unless otherwise indicated; *^b^* includes one patient who is still mechanically ventilated in the ICU; *^c^* Mann–Whitney U test and X^2^ with Yates correction or Fisher’s exact test comparing those who survived versus died in the ICU; *^d^* coronary artery disease and/or congestive heart failure; *^e^* lymphoma, chronic lymphoid leukemia and acute myeloid leukemia had 3,1 and 1 patients, respectively (3 patients were under chemotherapy); *^f^* defined as BMI > 30 kg/m^2^; *^g^* upper limit of normal 0.5 mg/dL; *^h^* upper limit of normal 0.1 ng/mL; *^i^* upper limit of normal 0.3 μg/mL; *^j^* upper limit of normal 14 pg/mL; *^k^* refers to 41 patients who received invasive mechanical ventilation. Bold: Indicates the presence of statistical significance.

**Table 2 jcm-09-03730-t002:** Interventions in the ICU and outcomes *^a^*.

	All (*n* = 50) *^b^*	Survived ICU (*n* = 33)	Died in ICU (*n* = 16)	*p ^c^*
High-flow nasal cannula	14 (28)	13 (39)	1 (6)	**0.019**
Non-invasive mechanical ventilation	2 (4)	2 (6)	0 (0)	0.551
Invasive mechanical ventilation (IMV)	41 (82)	24 (73)	16 (100)	**0.021**
Time to intubation, median (IQR) *^d^*	2 (0–3)	2 (0–4)	1 (0–3)	0.249
IMV days, median (IQR)	13 (9–33)	13 (10–32)	14 (7–27)	0.847
Neuromuscular blockade	26 (52)	14 (42)	12 (75)	0.066
Prone position	6 (12)	4 (12)	2 (13)	0.969
Noradrenaline	34 (68)	17 (52)	16 (100)	**<0.001**
Noradrenaline days, median (IQR)	6 (2–13)	4 (1–8)	11 (5–15)	**0.009**
Renal replacement therapy	13 (26)	4 (12)	8 (50)	**0.003**
Renal replacement therapy days				
median (IQR)	15 (11–24)	19 (15–24)	15 (9–26)	0.570
Selected inpatient medications				
Hydroxychloroquine	44 (88)	29 (88)	14 (88)	0.969
Azithromycin	38 (76)	26 (79)	12 (75)	0.765
Lopinavir/Ritonavir	18 (36)	13 (39)	5 (31)	0.811
Anti-Interleukin-6 antibody	5 (10)	3 (9)	2 (13)	0.893
Remdesivir (or placebo)	3 (6)	1 (3)	1 (6)	0.813
Glucocorticoids	5 (10)	2 (6)	3 (19)	0.382
Outcomes				
Follow-up, days (median (range))	50 (29–88)	50 (29–84)	50 (30–88)	NA
Successful extubation *^d,e^*	17 (41)	17 (71) *^f^*	0 (0)	NA
Time to successful extubation, days				
(median (IQR)) *^d,e^*	9 (5–14)	9 (5–14)	NA	
Tracheostomy *^d^*	12 (29)	7 (29) *^f^*	4 (25)	0.942
Time to tracheostomy, days				
(median (IQR)) *^d^*	22 (19–25)	21 (19–26)	22 (18–26)	0.968
Mechanical ventilation days,				
median (IQR) *^d^*	13 (9–33)	13 (10–32)	14 (7–27)	0.847
ICU days, median (IQR)	14 (9–31)	15 (10–30)	14 (7–27)	0.417
Hospital days, median (IQR)	21 (10–32)	24 (15–35)	14 (7–27)	**<0.001**
28-day mortality	12 (24)	NA	NA	NA
28-day mortality in ventilated *^d^*	12 (29)	NA	NA	NA

ICU, intensive care unit; IQR, interquartile range; NA, not applicable. *^a^* Data are expressed as *n* (%) unless otherwise indicated; *^b^* includes one patient who is still mechanically ventilated in the ICU; *^c^* Mann–Whitney U test and X^2^ with Yates correction or Fisher exact test comparing those who survived versus those who died in the ICU; *^d^* refers to 41 patients who received invasive mechanical ventilation; *^e^* among patients who did not have a tracheostomy; *^f^* patients who survived and were discharged alive from the ICU (*n* = 24) were the sum of patients who were successfully extubated (*n* = 17) and those who had a tracheostomy (*n* = 7). Bold: Indicates the presence of statistical significance.

**Table 3 jcm-09-03730-t003:** Comparison of intubated ARDS patients with COVID-19 with an historical control group of patients with ARDS without COVID-19 [16] *^a^*.

	ARDS with COVID-19 (*n* = 41)	ARDS without COVID-19 (*n* = 64)
Age, years	65.7 ± 10.4	52.9 ± 17.1
Gender, male	31 (75.6)	47 (73.4)
Comorbidity		
Hypertension	12 (29.3)	19 (29.7)
Diabetes mellitus	7 (17.1)	7 (10.9)
Malignancy	5 (12.2)	9 (14.1)
Other	5 (12.2)	6 (9.4)
APACHE II score	14.6 ± 6.9	19.3 ± 7.8
SOFA score	7.9 ± 3.3	12.1 ± 2.6
Hemodynamic variables and support		
MAP, mmHg	82.2 ± 6.7	77.3 ± 11.0
Lactate, mmol/L	1.7 ± 1.1	2.8 ± 2.8
Noradrenaline	33 (80)	50 (78)
Arterial blood gases		
PaO_2_, mmHg	115.8 ± 42.8	78.2 ± 12.7
PaO_2_/FiO_2_, mmHg	147.5 ± 74.5	106.9 ± 27.7
PaCO_2_, mmHg	42.6 ± 10.2	47.5 ± 8.0
pH	7.35 ± 0.09	7.30 ± 0.08
Respiratory parameters		
Tidal volume, L	0.47 ± 0.04	0.45 ± 0.06
Ventilation rate, breaths/min	24.3 ± 3.3	27.2 ± 5.3
PEEP, cmH_2_O	13.9 ± 3.7	13.1 ± 3.0
Plateau pressure, cmH_2_O	26.7 ± 3.5	29.9 ± 3.0
Driving pressure, cmH_2_O	12.8 ± 2.1	16.4 ± 2.3
Static compliance, mL/cmH_2_O	38.9 ± 9.3	28.8 ± 5.3
LIS	2.9 ± 0.5	3.4 ± 0.8
Prone position	6 (15)	12 (19)
Outcomes		
Follow-up, days (median (range))	50 (29–82)	65 (17–142)
ICU days (survivors), median (IQR)	14 (9–31)	19 (10–45)
Hospital days (survivors), median (IQR)	20 (8–30)	31 (12–55)
Hospital mortality	16 (39.0)	41 (64.1)

ICU, intensive care unit; IQR, interquartile range; APACHE, Acute Physiology and Chronic Health Evaluation [18]; SOFA, Sequential Organ Failure Assessment [19]; MAP, mean arterial pressure; PaO_2_, partial pressure of arterial oxygen; FiO_2_, fraction of inspired oxygen; PEEP, positive end-expiratory pressure; LIS, Lung Injury Score [20]. *^a^* Data are expressed as mean ± SD or *n* (%), unless otherwise indicated.

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
