# Peer review of "Hospital Resources May Be an Important Aspect of Mortality Rate among Critically Ill Patients with COVID-19: The Paradigm of Greece"

_jcm, 2020, doi:10.3390/jcm9113730_

Round 1
Reviewer 1 Report
Dear authors,
Thank you for providing research on such an important topic as coronavirus disease 2019 (COVID-19). The improvements in medicaments, treatment strategies, and hospitalization course should be done in order to minimize the number of potential side effects. Even though the concept of this work is very appealing, the presentation of the results could be improved.
- Firstly, the manuscript would benefit from English editing, there are several either minor or major grammatical mistakes that are quite misleading for the readers.
- When the authors start to use abbreviations such as ‘ARDS’ for example, please use it consistently thereafter. See line 53 for example. Please check the manuscript in terms of other abbreviations as well to unify it.
- Line 40 – please add references – see Baj et al. 2020
- The work would benefit from providing a control group which for example could be a group of patients with ARDS however without CO This might probably help in comparing the results thus, providing more specific results and conclusions. The comparison of your results with other authors is acceptable however, it should be included in the discussion section; the results from other studies cannot replace the control group which should be included in this study.
- Due to the different pharmacotherapy strategies, the results seem to be quite incomparable.
- No further radiological data is provided (the authors have only mentioned about the bilateral infiltrates with quantification of the lesions). It would be beneficial to present the radiological data as well.
- There is no data regarding any post-mortem studies. Is there any possibility to add such information?
Reviewer 2 Report
General Comments
Greece has been an outlier in the number of cases of COVID-19 and the mortality of affected patients. I was hoping this manuscript was going to shed some light on why Greece has had such a unique experience. Ref #11 (by a surgeon) provides some information, but its framework is simplistic and the analysis is superficial. I recognize that you have a narrower focus, but it would help readers to place your work in context by providing a reference containing a more cogent analysis (even if the source was not published in a medical journal) of why Greece has been such a COVID-19 outlier.
I find no major flaws with your data collection or analysis. The weakest part of the manuscript is the discussion. You have a unique data set with high internal validity: all your patients were admitted to a single hospital, you have the same staff, and had no shortage of resources. But your discussion does not offer any novel ideas or insights beyond what I have seen in dozens of other articles. Why do you want to publish this manuscript?
Specific Comments
Lines 62-64: You write: “Although there was an institutional recommendation against the routine use of non-invasive positive pressure mechanical ventilation, the routine use of high-flow nasal cannula oxygen therapy was not limited.”
Please provide additional information on this point. The sentence is written ambiguously. It could be interpreted that your institution discouraged the use of non-invasive ventilation even before the Covid pandemic.
If the institutional recommendation against the routine use of non-invasive ventilation did not happen until after the onset of Covid, what was the basis of the institutional recommendation?
Was it based on WHO Guidelines or Surviving Sepsis Guidelines, which did discourage use of non-invasive ventilation?
The present sentence is ambiguous and it needs to be clarified for readers.
Lines 102 …: Table 1: The numbers in the 2 columns do not add up. All were 50, 33 survived, and 16 died. Where is the missing patient?
Page 4 Table 1: For ABGs, you provide PaO2/FiO2 and also PaCO2, but you omit PaO2 values. You need to list PaO2 values.
PaO2/FiO2 values are intrinsically unreliable because it is highly problematic to get accurate FiO2 values in a non-intubated patient.
Moreover, severity of hypoxemia is of particular interest in Covid patients, and the carotid bodies do not respond to PaO2/FiO2.
Line 135, a: You write: “Invasive mechanical ventilation was applied in 41 patients (82%).”
Do you know who made the decision to intubate the 41 patients? I assume the reasons are not documented in your medical record (typically they are not). Yet if you do know the reasons for the intubations, these should be reported.
Line 135, b: What was the management in the 9 patients who were not intubated?
Did the 9 patients receive any form of oxygen or any form of noninvasive ventilation?
How much concern was there about exposure to the virus among the staff?
Line 135, c: Among the 41 intubated patients, how many were first tried on noninvasive ventilation (give the precise number) or on high flow nasal cannula (give precise numbers)?
Lines 171-172: You write: “among patients who died, 3 (19%) had suffered cardiac arrest during endotracheal intubation.”
Did any of the survivors experience a cardiac arrest during intubation?
Lines 172-173 a: You write: “Compared to those who survived, patients who died in the ICU rarely received high-flow nasal cannula oxygen therapy, and more commonly required invasive mechanical ventilation,..”
Your choice of the two verbs in this sentence – “received” and “required” – is most insightful. Why do you say that patients getting high-flow nasal cannula oxygen therapy “received” it whereas patients who were mechanically ventilated required it?
I recommend that you read the paragraph on circular reasoning in the following editorial [PMID: 32281885] and also other concepts in the editorial that are pertinent your study:
https://www.atsjournals.org/doi/pdf/10.1164/rccm.202004-1076ED
Lines 172-173 b: I recognize that common sense would suggest that sicker patients are more likely to receive invasive ventilation than less sick patients – but there can be exceptions.
You have measure of the degree of sickness of the patients who are ventilated and in the patients that were non-ventilated.
You should inform readers whether or not these measures were different in the two groups of patients.
Lines 205-206 a: You write: “ICU and hospital mortality were 32% and 39% for the entire cohort of patients admitted to the ICU and for those who received invasive mechanical ventilation, respectively.”
While it is okay to inform readers of this information, it would be more interesting to inform readers of the mortality in the patients who are intubated versus the mortality in the patients who were not intubated.
Is there a statistically significant difference in the mortality in the intubated versus the non-intubated groups or were your numbers too small to achieve statistical significance?
Among patients who were reasonably young (say under 60-65 years) and without significant co-morbidity, what was the mortality in the patients who were intubated versus in the patients who were not intubated?
Lines 215-236: This entire paragraph is disappointing. I was hoping you were going to come up with something imaginative and creative in generating lessons from your data set. Instead the entire paragraph reads as if the primary goal of researchers in Greece was to see if they could reproduce findings reported by investigators in the state of Georgia.
The report from Georgia is certainly of interest, but do you think the findings are so novel, so surprising, and so important for structuring medical care that they are in need of confirmation?
Do you think that the similarity in the numerical values between Greece and Georgia convey a major biological fact or are the result of coincidence?
Do you have any concern about duplicate publication between two papers from the Georgia group [PMID: 32452888 and PMID: 32804790]?
Lines 227-236 a: Much of this section is devoted to a discussion of ARDS, but I have difficulty in finding novelty in the statements.
You convey that the mortality with Covid ARDS in Greece is similar to reported mortality for ARDS in general. If so, has the large number of papers addressing the question of whether Covid patients have Typical ARDS or Atypical ARDS been a distraction? See PMID: 32707184
https://journal.chestnet.org/article/S0012-3692(20)31957-7/fulltext
Lines 227-236 b: You present several mortality percentages. Do you believe these to have biological significance or are they merely sociological observations?
Lines 237-243: The mortality rate that you observed is distinctly different from the mortalities reported in several other studies. Yet the sentences here are not illuminating.
Is there anything distinctive about your mortality rate?
Do you have an explanation for why your mortality rate differs from that of other groups?
Lines 256-257: You write: “the same ventilatory protocol was also applied in both groups as implied by the presence of no difference in PEEP, plateau pressure and driving pressure levels.”
There are problems with elementary logic: once again, you are engaging in circular reasoning.
The lack of differences in PEEP, plateau pressure and driving pressure does not signify that a protocol was followed. You may have developed a protocol, but it is a very different thing to prove that people were actually employing it.
Lines 275-282: Your conclusions are weak, and they do not distinguish your article from the dozens of other articles that have been published on COVID-19.
Lines 280-282: Your final sentence is a non sequitur. How could mortality rate for an overall group and then a mortality rate for patients who received mechanical ventilation serve as the basis for future treatment protocols? Why would this be a worthwhile desideratum? No part of your study was conducted to determine the efficacy of using protocols.
Title of your manuscript: The title of your manuscript is uninviting.
It will not attract the attention of many readers.
You should mention the word Greece in the title because people will want to know why Greece was such an outlier.
Round 2
Reviewer 1 Report
Dear Authors
Please correct the figures according to the guidelines for the authors. Please check the guidelines for the authors and correct it according to it. This suggestion is primarily for figure 1. Both of the tables should have separate titles.
Except for this minor correction, all of the previous comments and suggestions were corrected in a proper manner.
Thank you for your contribution that might constitute a basis for further improvements in the management with COVID-19 patients which is of great importance right now.
Reviewer 2 Report
General Comments
The manuscript is much improved.
There are still a number of important problems that need to be addressed. These relate to problems with the choice of words in the English language and also to information that you provided in your point by point response cover letter but failed to include in the revised manuscript.
Line 4: the title refers to “adequate” hospital resources. A better choice would be “sufficient” rather than “adequate.”
Line 35: “acceptable mortality” is something of an oxymoron. I suggest you rewrite it as “understandable mortality” or some other synonym.
Line 57: On this line, and a number of other places you referred to “overwhelming” hospital resources. I think it would be better if you talk about “overburdened” staff.
Line 76-78: you state that was us “an institutional recommendation against the routine use of non-invasive positive pressure ventilation…”. In your point by point response cover letter, you write: “It was mainly based on the Surviving Sepsis Campaign Guidelines.” It is important to insert this information – It was mainly based on the Surviving Sepsis Campaign Guidelines – into the revised manuscript.
Line 158-167: you provide additional information concerning the patients who are intubated and placed on mechanical ventilation – this is good. In the revised manuscript I do not see details about the management of the patients who are not intubated. In R5 of your point by point response letter, you write: “All these 9 patients were managed with high-flow nasal cannula oxygen therapy with 60 L/min air-flow and high FiO2; two of these patients were also managed with noninvasive mechanical ventilation.” It is important to insert this information into the revised manuscript.
Line 299: you refer to inadequate hospital resources – it would be more precise to speak of constrained hospital resources or insufficient hospital resources.
Line 307: you refer to “overwhelmed” – it would be more precise to speak of “overburdened”
